# Increased male investment in sperm competition results in reduced maintenance of gametes

**Mareike Koppik[1,2], Julian Baur[1], David Berger[1] \***

**1** Department of Ecology and Genetics, Animal Ecology, Uppsala University, Uppsala, Sweden,
**2** Department of Zoology, Animal Ecology, Martin-Luther University Halle-Wittenberg, Halle (Saale), Germany

\* david.berger@ebc.uu.se

## Abstract

Male animals often show higher mutation rates than their female conspecifics. A hypothesis for this male bias is that competition over fertilization of female gametes leads to increased male investment into reproduction at the expense of maintenance and repair, resulting in a trade-off between male success in sperm competition and offspring quality. Here, we provide evidence for this hypothesis by harnessing the power of experimental evolution to study effects of sexual selection on the male germline in the seed beetle *Callosobruchus maculatus*.

We first show that 50 generations of evolution under strong sexual selection, coupled with experimental removal of natural selection, resulted in males that are more successful in sperm competition. We then show that these males produce progeny of lower quality if engaging in sociosexual interactions prior to being challenged to surveil and repair experimentally induced damage in their germline and that the presence of male competitors alone can be enough to elicit this response. We identify 18 candidate genes that showed differential expression in response to the induced germline damage, with several of these previously implicated in processes associated with DNA repair and cellular maintenance. These genes also showed significant expression changes across sociosexual treatments of fathers and predicted the reduction in quality of their offspring, with expression of one gene also being strongly correlated to male sperm competition success. Sex differences in expression of the same 18 genes indicate a substantially higher female investment in germline maintenance.

While more work is needed to detail the exact molecular underpinnings of our results, our findings provide rare experimental evidence for a trade-off between male success in sperm competition and germline maintenance. This suggests that sex differences in the relative strengths of sexual and natural selection are causally linked to male mutation bias. The tenet advocated here, that the allocation decisions of an individual can affect plasticity of its germline and the resulting genetic quality of subsequent generations, has several interesting implications for mate choice processes.

**Data Availability Statement:** The data sets supporting the conclusions of this article are available as follows: Gene expression data (raw reads and gene counts) of the main experiment (germline maintenance) are available at GEO under

accession number GSE153232; phenotypic data of the sperm competition and germline maintenance experiment as well as normalized log2 counts of the 18 irradiation responsive genes for the experimental evolution lines are available at Mendeley Data (http://dx.doi.org/10.17632/szyb2z8rzk.1).

**Funding:** This work was financially supported by Vetenskapsrådet (https://www.vr.se, grant no. 2019-05024 to DB), by Carl Tryggers Stiftelse för Vetenskaplig Forskning (https://www.carltryggersstiftelse.se, grant no. CTS18:32 to DB), and by Fysiografiska sällskapet i Lund (https://www.fysiografen.se, "Nilsson-Ehle" to MK). The financial funders had no role in study design, data collection and analysis, decision to publish, or preparation of the manuscript.

**Competing interests:** The authors have declared that no competing interests exist.

# Background

The germline mutation rate impacts on a range of evolutionary processes such as the rate of adaption [1,2] and risk of extinction [3,4], as well as genome evolution [5]. Contrary to the typical assumption in population genetic models, recent studies have shown that mutation rate can vary both within and between individuals within a given species [6–16]. Such variability can, for example, affect genetic load at mutation–selection balance [17,18], mate choice and the efficacy of sexual selection [7], and adaptation in stressful environments [8,19,20]. Despite these important implications, experimental evidence providing ultimate causation for the observed intraspecific variability in mutation rates remains scarce [21–23].

One prominent type of intraspecific variation is that between males and females. Males often show higher germline mutation rates in animal taxa [24–28], including humans [29,30] and other primates [22,28]. This male mutation bias has been ascribed to the greater number of cell divisions occurring in the male germline prior to fertilization, and the higher number of divisions in males is itself thought to be a result of anisogamy and sexual selection promoting increased gamete production in the sex competing most intensively for fertilization success [31,32]. Indeed, a need for fast-dividing male germline cells would inevitably lead to an elevated risk of unrepaired replication errors in male gametes, all else equal, as the DNA-repair system must constantly attend single and double strand breaks that occur during meiosis and mitosis [33–36] and in post-meiotic chromatin remodelling during spermiogenesis [29,36,37]. This should result in a trade-off between increased male germline replication rates, granting greater success in sperm competition, and increased germline mutation rate, reducing offspring quality [7,33,38–41]. However, whether there generally is abundant variation in germline replication rates within natural populations, how such variation relates to the realized mutation rate, and what role sperm competition plays in shaping this variation, remains largely unknown.

Several lines of evidence suggest that germline replication rate is not the only factor affecting mutation rate. For instance, sex differences in the number of germline cell divisions do not perfectly predict male mutation bias across species [22,25,27,42,43], and in humans, differences in mutation rate between males of the same age can be many times greater than that between the sexes [12,13,30]. Both points suggest that maintenance processes may be central in deciding the germline mutation rate both within and across species. Maintenance of the germline is energetically costly [36,37,44] and comprises interrelated processes such as antioxidant defence [45–47], repair of DNA damage [20,34], and programmed cell death of damaged sperm [48]. Indeed, the male gonad is a highly oxidative environment that, without antioxidant defence devoted to dealing with reactive oxygen species [41,45–48], produces DNA damage that may result in germline mutations [35–37,49]. Recent studies imply that costly male allocation decisions involving sperm and ejaculate production and composition [50–56] are responsive to female characteristics such as mating status [57] and to the presence of conspecific males [52,58], most likely because these serve as cues for predicting mating opportunities and the level of competition a male's sperm may encounter [59]. Thus, sociosexual cues that signal increased risk of sperm competition and increase allocation to ejaculate components that increase male post-copulatory reproductive success could lead to concomitant plastic decreases in germline maintenance and reduced gamete quality. However, direct experimental evidence supporting this hypothesis remains very scarce indeed.

Here, we tested this prediction using experimental evolution lines of the seed beetle *Callosobruchus maculatus*, a model organism for sexual selection where sperm competition is rife [7,60–64]. These lines have been maintained for >50 generations under 3 alternative mating regimes manipulating the relative strength of natural and sexual selection: natural polygamy

applying both natural and sexual selection (N+S regime), enforced monogamy applying natural selection while removing sexual selection completely (N regime), or a sex-limited middle class neighborhood breeding design [65,66] applying sexual selection while minimizing natural selection on fecundity and juvenile viability (S regime). The S regime does not show any strong signs of decline in fitness-related traits when assayed at standard conditions [67,68]. However, previous findings suggest that S males pass on a greater genetic load to their progeny if having engaged in sociosexual interactions prior to being challenged with a dose of irradiation introducing DNA damage in their germline, an evolved response to sociosexual interactions not seen in N or N+S males [7]. A plausible explanation for this result is that S males have evolved reduced germline maintenance as a response to increased post-copulatory sexual selection coupled with weakened constraints on the evolution of sperm and ejaculate traits in this mating regime, where viability selection was minimized.

In order to test this hypothesis, we first determined sperm competitiveness in males of all experimental evolution regimes to confirm that S males indeed evolved adaptations to post-copulatory sexual selection. Subsequently, we focused on the sociosexual effect on germline maintenance in S males. Here, we set out to determine if the presence of conspecific males (increasing competition) and females (mating opportunities) triggered the change in germline maintenance in S males. Additionally, we employed RNA sequencing of the reproductive tracts of S males to gain insight into the possible mechanisms behind this change. In a last step, we compared the expression of irradiation responsive genes in males and females from all experimental evolution regimes to quantify putative sex differences in germline maintenance.

## Results

### Sperm competition

To test whether strong sexual selection coupled with the removal of natural selection in S males had led to the evolution of traits ensuring higher post-copulatory reproductive success, we assayed male sperm competition success in defense (P1: focal male is first to mate) and offense (P2: focal male is second to mate) for all 3 evolution regimes; 3 lines for N and N+S and 2 lines for the S regime (1 line was accidentally lost during the experimental evolution). The fact that we only have 2 replicates for the S regime warrants some caution when interpreting results. However, these 2 lines tend to behave very similarly in our experiments (see S1 Appendix for sperm competition results). Furthermore, the statistical methods used take into account that data come from only 2 replicated S lines and should represent rather conservative estimates of statistical significance. In order to estimate sperm competition success of males, females from the ancestral population, from which the experimental evolution lines were derived, were mated twice (once to a focal male and once to a competitor) with 24 h in between matings, during which time the females were provided with beans for egg laying. The competitor males were from a black mutant strain [69] such that paternity in offspring could be determined (Fig 1A). Focal males were either held singly ("solitary" treatment) or in groups of 5 males from the same line prior to mating ("competition" treatment). For P2, males were also tested in their first, third, and last of 5 consecutive matings to determine effects of sperm and seminal fluid depletion.

For P1, there was no clear difference between regimes averaged across the 2 social contexts (solitary or competition). However, there was a strong tendency for S males and N males to differ in their responses to social context (interaction: $P_{MCMC} = 0.052$). When held as solitary, males of the 2 replicate S lines tended to have higher P1 than males of the other lines (Fig 1B and S1 Appendix), although differences between regimes were not statistically significant (S versus N: $P_{MCMC} = 0.098$, S versus N+S: $P_{MCMC} = 0.072$). When experiencing competition, P1

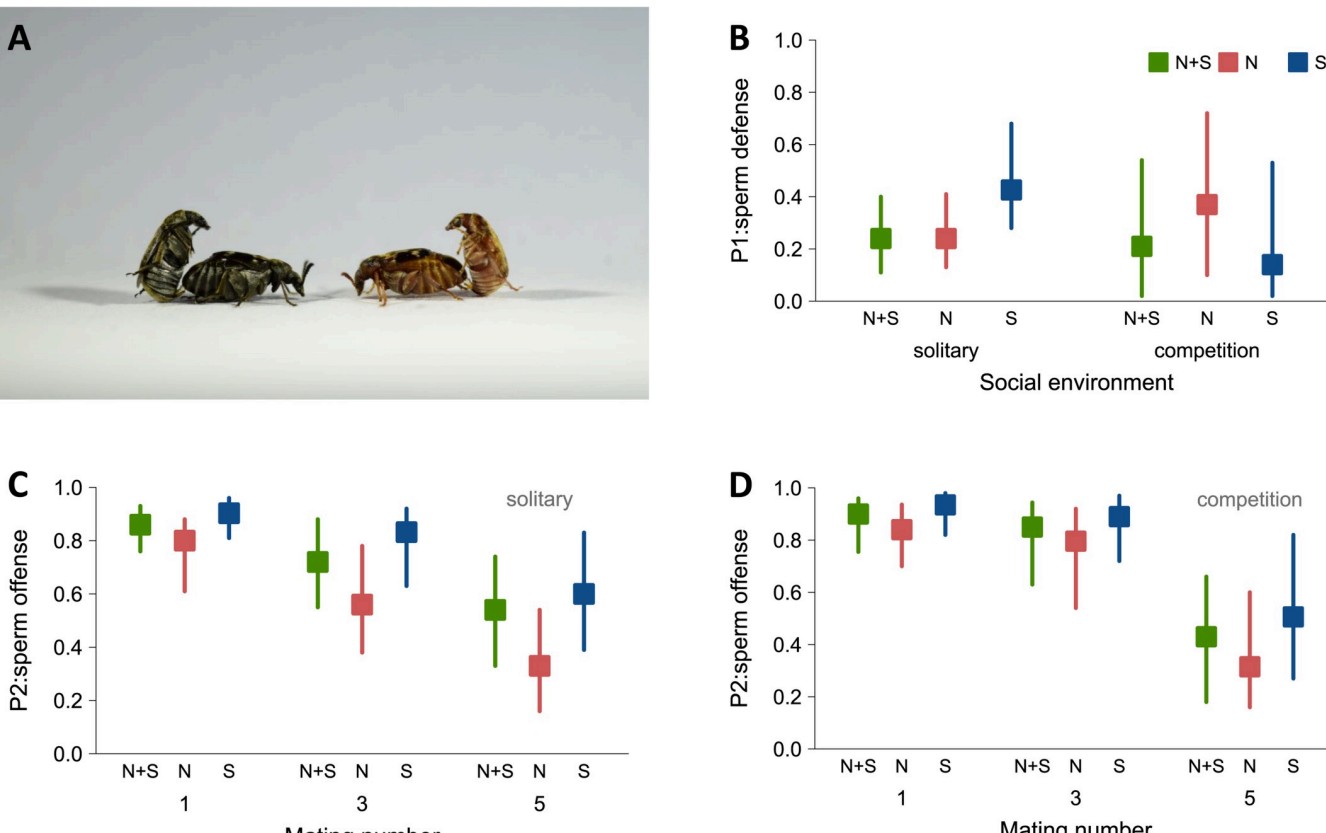

**Fig 1. Sperm competition success in males from the experimental evolution lines.** (A) Paternity was estimated by competing a standard male deriving from a black strain (left) to focal males of each regime (right). In (B), paternity share when the focal male was first to mate (P1). In (C) and (D), paternity share when the focal male was second to mate (P2). S (blue), N (red), and N+S (green). Shown are posterior modes and 95% credible intervals from Bayesian mixed effect models. The data underlying this figure can be found in "SpermCompetition.txt" at Mendeley Data (http://dx.doi.org/10.17632/szyb2z8rzk.1).

of S males tended to decline, and, if anything, there was a tendency for higher P1 in N males in this social context (Fig 1B). However, the replicate lines of the N regime showed very different responses to social context (S1 Appendix), and there were no significant differences between S males and the other regimes in the competition treatment (S versus N: $P_{MCMC} = 0.27$, S versus N+S: $P_{MCMC} = 0.80$). There are at least two potentially simultaneously acting processes that may explain the difference in how P1 responded to social context in S and N males. First, male–male competition is very costly in *C. maculatus* [70] and potentially more intense in S males than in N males [7,71], which may have impaired P1 of S males under competition (see Discussion). Second, the presence of male competitors differentially shapes both sperm and ejaculate production in S versus N males [7], which may indicate changes to the post-copulatory strategy of competing S males. In line with this argument, the S regime had consistently higher P2 than the N regime across contexts ($P_{MCMC} = 0.012$; Fig 1C and 1D). S males did not have significantly higher P2 than N+S males ($P_{MCMC} = 0.19$; Fig 1C and 1D). There was no effect of social context on P2 (all $P_{MCMC}$ including social context > 0.6; Fig 1C and 1D). P2 declined in successive matings, suggesting ejaculate depletion (Mating 1 versus 5: $P_{MCMC} < 0.001$, Fig 1C and 1D), but there were no significant differences between the 3 regimes in how successive mating affected P2 (all interactions: $P_{MCMC} > 0.14$; S1 Appendix). Finally, we performed a global model in which P1 and P2 (including all social contexts) were analyzed together. This model showed that S males had a significantly higher overall sperm competition

success compared to N males across our experiments ($P_{MCMC}$ = 0.012), and also, albeit nonsignificant, a tendency for higher success compared to N+S males ($P_{MCMC}$ = 0.088) (Fig 1; for model summaries, see S1 Appendix).

### Germline maintenance

**Offspring quality.** Since S males have evolved enhanced post-copulatory competitiveness, we hypothesized that they invest more into mating and competition than N and N+S males at a potential cost of reduced germline maintenance. Indeed, S males have previously been shown to reduce germline maintenance when engaging in inter-and intrasexual interactions with conspecifics (Fig 2A and Baur and Berger (2020) [7]). To dissect the effects of inter- and intrasexual interactions on germline maintenance, we manipulated the social environment of S males in a full-factorial design (with or without male competitors and with or without female mating partners; Fig 2B). We did not control for matings in the groups with intersexual interactions. However, by keeping the male-to-female ratio the same in groups with and without competitors, we expect that the average number of copulations per male is approximately the same in the two groups, and we mainly introduced a higher variance between individual males in the groups with competitors present. We then measured the reduction in offspring quality for those males after a short (approximately 3 h) and long (approximately 24 h) recovery period following the induction of germline damage through gamma radiation. To this end, we mated males to a single virgin female at each time point (3 h and 24 h post-irradiation treatment) and established a second generation from the resulting offspring. This allowed us to estimate the quality of offspring from F0 irradiated fathers by counting the number of F2 progeny produced in those lineages relative to F2 progeny production in lineages deriving from unirradiated F0 control males. Hence, the reduction in offspring quality could be calculated as: 1-[$F2_{IRRADIATED}$/$F2_{CONTROL}$].

In agreement with previous results [7], males in the mixed treatment, including both male–male and male–female (mating) interactions, fathered offspring of lower quality in both matings and for both lines when challenged to repair damage in their germline (Fig 3). Moreover, we found that male–male interactions decreased offspring quality after the short-term 3-h recovery period (Irradiation × Intrasexual interactions: $P_{MCMC}$ = 0.010; Fig 3A). A similar, albeit nonsignificant, effect was found for mating interactions (Irradiation × Intersexual interactions: $P_{MCMC}$ = 0.088; Fig 3A). After the long-term recovery period, with males being held in isolation during the 24 h post-irradiation resting period, the effect of male–male interactions was no longer detectable (Irradiation × Intrasexual interactions: $P_{MCMC}$ = 0.844; Fig 3B), while the effect of previous matings persisted (Irradiation × Intersexual interactions: $P_{MCMC}$ = 0.030; Fig 3B). The 2 lines differed overall in the observed reduction in offspring quality but showed similar responses to the sociosexual treatments (Fig 3; for model summaries, see S2 Appendix).

**Differential gene expression.** To explore the molecular underpinnings of the reduction in germline maintenance, we took a subset of males from all treatment groups (line/irradiation/social environment) to analyze gene expression in the male reproductive tracts (Fig 2C) after short-term recovery. Analyzing all 12,161 genes expressed in our data set, we found strong differences between the 2 experimental evolution lines (5,910 differentially expressed genes (DEGs), 49%; Table 1) reflecting that these lines have been evolving separately for more than 50 generations and were reared in separate jars prior to the experiment. When looking at the effect of inter- and intrasexual interactions, most differences occurred in the environment that combined both types of interactions (mixed versus solitary, 3,418 DEGs, 28%; Table 1), with the effect of mating (mated versus solitary, 2,747 DEGs, 23%; Table 1) contributing more

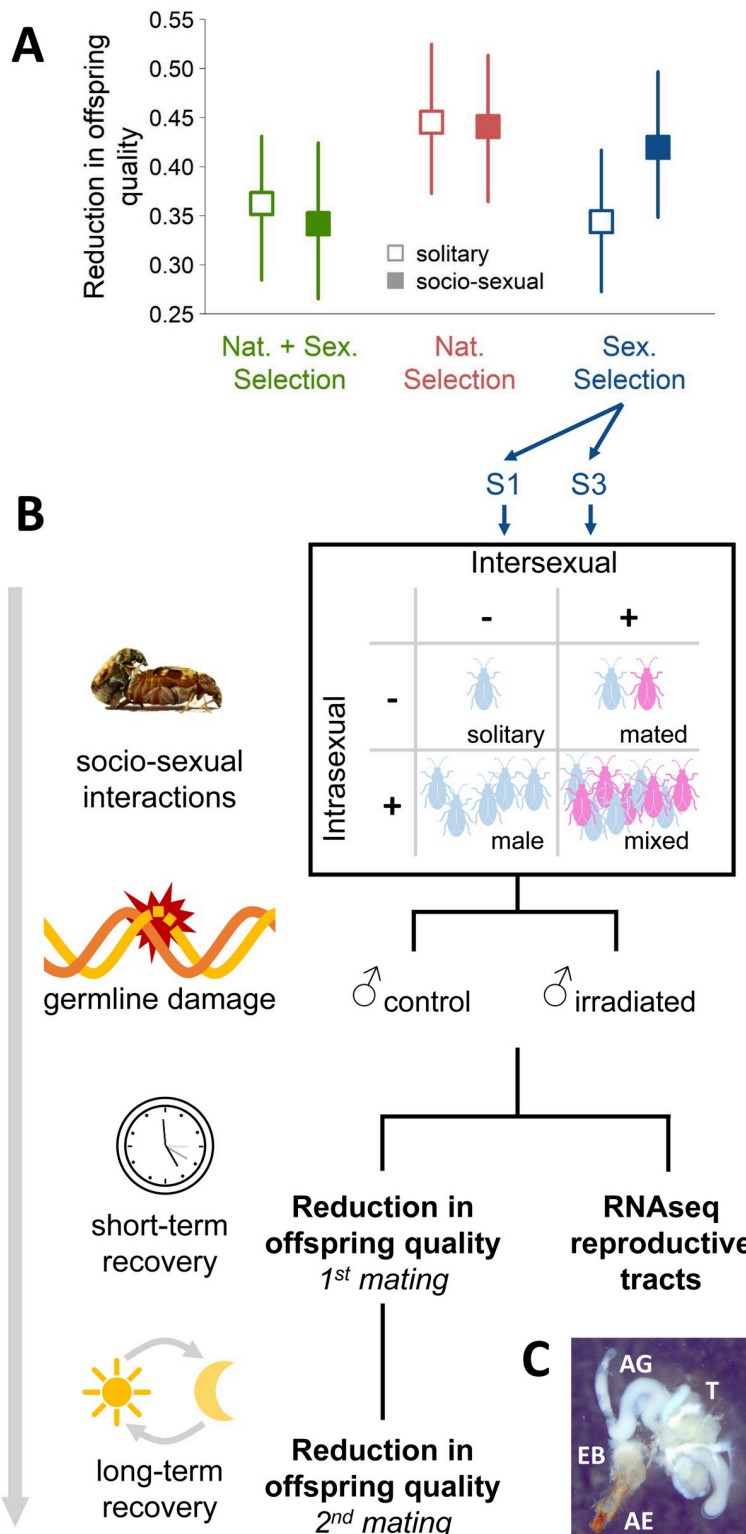

**Fig 2. Estimating germline maintenance.** (**A**) Reduction in offspring quality after induction of germline DNA damage through irradiation of male beetles. Solitary males (open symbols) with an evolutionary history of sexual selection (N+S and S males) suffer less reduction in offspring quality than males from lines with only natural selection acting (N males). However, sociosexual interactions prior to the challenge to the germline decreases offspring quality further in S males but not in N+S and N males (closed symbols). Data represent Bayesian posterior means and 95%

credible intervals from Baur and Berger (2020) [7]. Note that "offspring quality" here captures quality of both the F1 (fertility and fecundity) and F2 (juvenile-to-adult survival) generation. (**B**) Schematic overview of the experiment estimating germline maintenance. Males from 2 S lines were exposed to one of 4 sociosexual environments, manipulating the presence of conspecific males and females. Afterwards, we induced germline damage via gamma radiation and determined reduction in quality of offspring produced by those males after a short (approximately 3 h) and long (approximately 24 h) recovery period. Additionally, we examined gene expression in male reproductive tracts at the end of the short recovery period. (**C**) Picture of a male reproductive tract. In *C. maculatus*, the male reproductive tract consists of the aedagus (AE), ejaculatory bulb (EB), 5 accessory gland (AG) pairs (2 large and 3 small AG pairs), and a pair of bilobed testes (T). For the gene expression, the two large AG pairs were not included.

than the effect of male–male competition (male versus solitary, 2 DEGs, <1%; Table 1). Irradiation resulted in only very few gene expression changes across both lines and all social environments (irradiated versus control, 18 DEGs, <1%; Table 1), and only one of those showed a larger than 2-fold change (Fig 4B and S3 Appendix), which may, in part, be due to the timing of the measurements. Nevertheless, several of the DEGs are implicated in processes associated with germline maintenance and DNA repair. Among the up-regulated genes, 3 (*CALMAC_LOCUS18783*, *CALMAC_LOCUS9511*, and *CALMAC_LOCUS2860*) code for proteins containing an MADF domain [72], which can also be found in the putative transcription factor s*tonewall* in *Drosophila melanogaster*, which is involved in female germline stem cell maintenance [73]. Also up-regulated is the gene *CALMAC_LOCUS8201*, whose protein product contains a ULP protease domain [72]; in yeast, Ulp1 is involved in the sumoylation dynamics that play a critical role in the DNA damage response, specifically in the repair of double-strand breaks [74]. Among the down-regulated genes, *CALMAC_LOCUS9612* codes for a protein

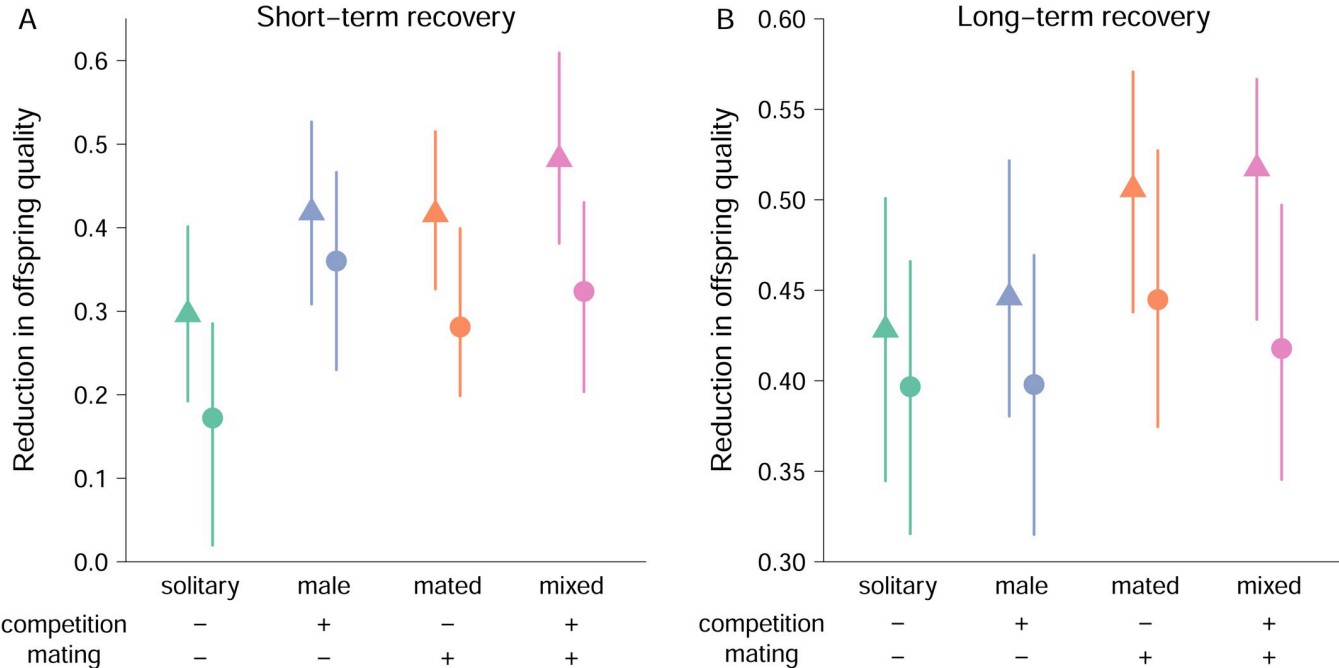

**Fig 3. Estimates of germline maintenance in males with an evolutionary history of intense sexual selection.** Reduction in offspring quality (1-[F2$_{IRRADIATED}$/F2$_{CONTROL}$]) after short-term (**A**) or long-term (**B**) recovery of males from 2 S lines (S1: triangles, S3: circles). Males were held in one of 4 different social environments before irradiation: solitary, without any competitors or mating partners (solitary, green symbols); without mating partners but with 4 male competitors (male, blue symbols); without competitors but with 1 female mating partner (mated, orange symbols); or with 4 male competitors and 5 female mating partners (mixed, pink symbols). Values are Bayesian posterior means and 95% credible intervals. The data underlying this figure can be found in "F2_short.txt" (**A**) and "F2_long.txt" (**B**) at Mendeley Data (http://dx.doi.org/10.17632/szyb2z8rzk.1).

**Table 1. Summary of differential gene expression analysis for the contrasts of interest, significance at 5% false discovery rate.**

|  | Line (MA3 –MA1) | Irradiation (irrad.—control) | Male (male—solitary) | Mated (mated—solitary) | Mixed (mixed—solitary) |
|---|---|---|---|---|---|
| down | 2,867 | 8 | 2 | 1,440 | 1,720 |
| not sign. | 6,251 | 12,143 | 12,159 | 9,414 | 8,743 |
| up | 3,043 | 10 | 0 | 1,307 | 1,698 |

containing a BIR domain [72], which can also be found in the *D. melanogaster* apoptosis inhibitors Iap1 and Iap2 [75].

Moreover, there was an overlap between genes responding to irradiation and to the social environments (specifically those treatments including intersexual interactions; S3 Appendix).

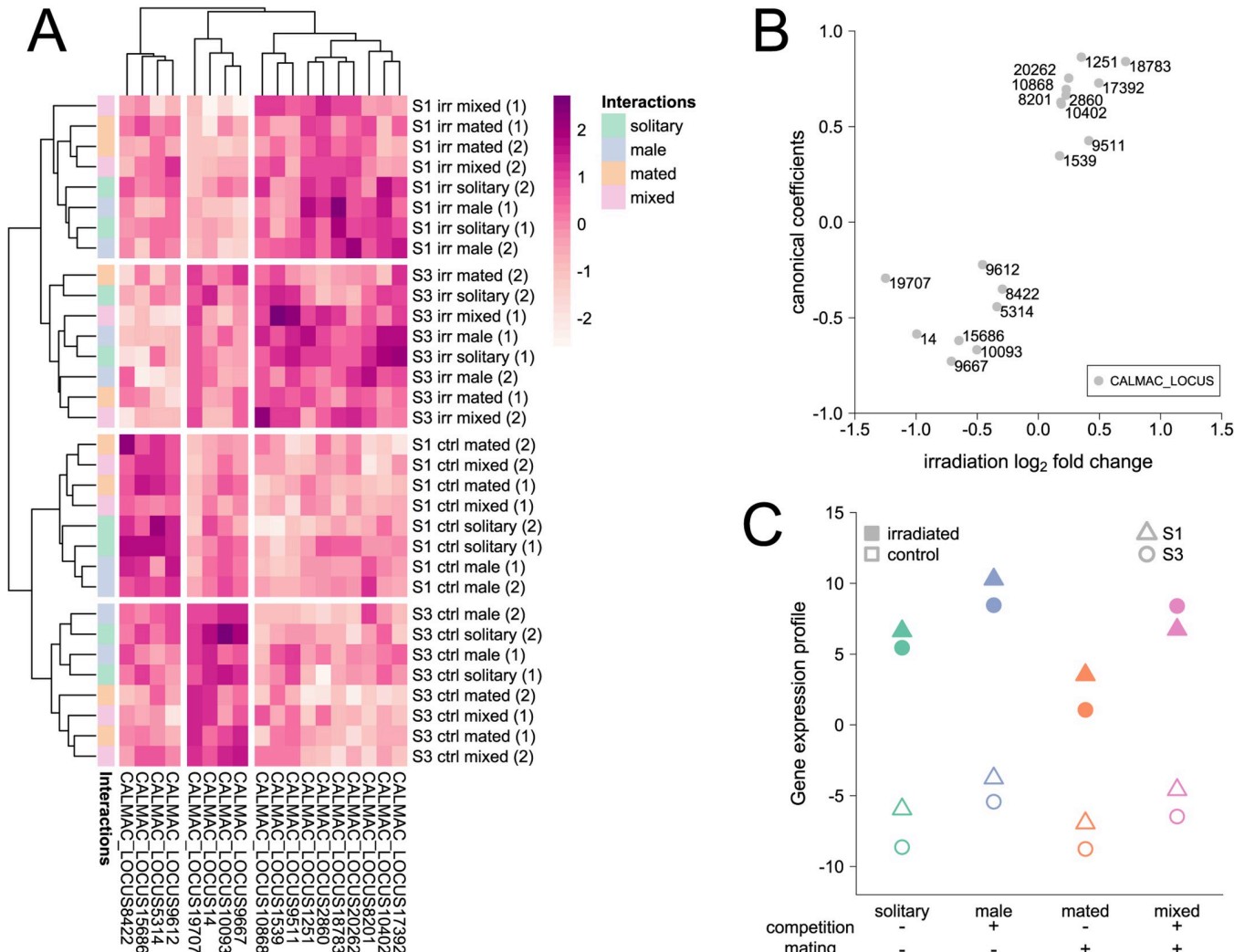

**Fig 4. Gene expression in the reproductive tracts of S males.** (**A**) Heatmap of scaled normalized $\log_2$ expression of the 18 genes that responded to the irradiation treatment. Expression is clearly separated between irradiation treatments (ctrl: control, irr: irradiated) and experimental evolution lines (S1, S3). Within these blocks, a separation between mated (orange and pink) and nonmated (green and blue) males can be observed. (**B**) Canonical coefficients of the 18 irradiation responsive genes (that make up the canonical scores of each sample) against their $\log_2$ fold change in response to irradiation. (**C**) Canonical scores separating control (open) and irradiated (closed) samples based on expression of the 18 irradiation responsive genes (triangles: S1; circles: S3). The data underlying this figure can be found at GEO under accession number GSE153232.

**Table 2. Test statistics of a multivariate analysis of variance with all 18 irradiation response genes as dependent variables.**

|  | df | Pillai's trace | *P* |
|---|---|---|---|
| **Line** | 1 | 0.98861 | **<0.001** |
| **Irradiation** | 1 | 0.99632 | **<0.001** |
| **Inter(sexual interactions)** | 1 | 0.98812 | **<0.001** |
| **Intra(sexual interactions)** | 1 | 0.93388 | **0.032** |
| **Irradiation × Inter** | 1 | 0.96718 | **0.005** |
| **Irradiation × Intra** | 1 | 0.62005 | 0.852 |
| **Inter × Intra** | 1 | 0.70972 | 0.662 |
| **Irradiation × Inter × Intra** | 1 | 0.77902 | 0.452 |

To explore effects of the social environment on irradiation responsive genes, we tested the 18 irradiation response candidate genes (Fig 4A) in a MANOVA. Here, we took advantage of our full-factorial design and tested the interaction between intersexual interactions, intrasexual interactions, and irradiation (Table 2). Both inter- and intrasexual interactions influenced overall expression of irradiation responsive genes independently (Table 2). Furthermore, intersexual interactions even affected the irradiation response itself (Table 2).

To further explore this link, we first conducted a canonical discriminant analysis to find a linear combination of expression values of irradiation responsive genes that best separates the irradiation and control samples. To get the best representation of the irradiation effect while avoiding overfitting the data, we controlled for variation due to line, social environment, and day and limited our interpretation to the first canonical axis. As expected, canonical coefficients for the 18 candidate genes roughly followed the $\log_2$ fold change induced by irradiation (Fig 4B). This graphical separation of irradiated from control samples recapitulated the statistical differences found between the social environments in the MANOVA (Fig 4C). For example, males cohabitated with other males (intrasexual interactions) already have slightly more "irradiation-like" canonical scores in control groups, thus without any germline damage induced. This may indicate that they already have an increased need for germline maintenance due to elevated investment in germline replication or ejaculate composition in response to competition, or in response to direct stress induced by male rivals, as male–male competition is pronounced and carries considerable costs in this species (e.g., [70]).

To determine whether the gene expression profiles of fathers from our treatment groups predicted the observed reduction in quality of offspring caused by the induced germline damage, we applied a canonical correlation analysis. Using the 2 lines and 4 sociosexual treatments as units of replication, the gene expression canonical scores of control and irradiated F0 fathers (Fig 4C) were entered as x variables, and the observed reduction in offspring quality after the short and long recovery period (Fig 3) as y variables. This thus resulted in 8 independent samples with 2 explanatory (gene expression) and 2 response (reduction in offspring quality) variables. Using F-approximations of Pillai–Bartlett's trace both canonical dimensions taken together are significant (F = 7.10, df1 = 4, df2 = 10, *P* = 0.006) with the second dimension also being significant on its own (F = 6.37, df1 = 1, df2 = 14, *P* = 0.024). Based on canonical dimension 1, more irradiation-like gene expression in control males was associated with greater reduction in the quality of offspring produced by the first mating following short-term recovery (Table 3). The gene expression of control males, not themselves being exposed to irradiation, likely reflects the state of the germline as a result of the sociosexual interactions, suggesting that the observed correlation between gene expression in the reproductive tract of the fathers and the reduction in quality of their

**Table 3. Canonical correlation coefficients.**

|  | Canonical coefficients | | Standardized coefficients | |
| --- | --- | --- | --- | --- |
|  | **Dimension 1** (corr.: 0.92) | **Dimension 2** (corr.: 0.79) | **Dimension 1** (corr.: 0.92) | **Dimension 2** (corr.: 0.79) |
| **Gene expression** | | | | |
| Control samples | −0.83 | −0.39 | −1.48 | −0.70 |
| Irradiated samples | 0.25 | 0.49 | 0.74 | 1.46 |
| **Reduction in offspring quality** | | | | |
| Short-term recovery | −11.71 | 10.38 | −1.14 | 1.01 |
| Long-term recovery | 4.27 | −33.16 | 0.19 | −1.51 |

offspring was driven by male–male interactions (Fig 5A). Canonical dimension 2 describes a correlation between the reduction in quality of offspring produced by the second mating following long-term recovery and the magnitude of the gene expression response to irradiation found in fathers (Table 3). This thus suggests that offspring quality is dependent on the capacity of fathers to modulate gene expression to deal with the induced damage, with stronger responses mitigating the consequences of germline damage. Here, the mating treatment, rather than male–male interactions, was the main driver of the correlation between germline gene expression and offspring quality (Fig 5B).

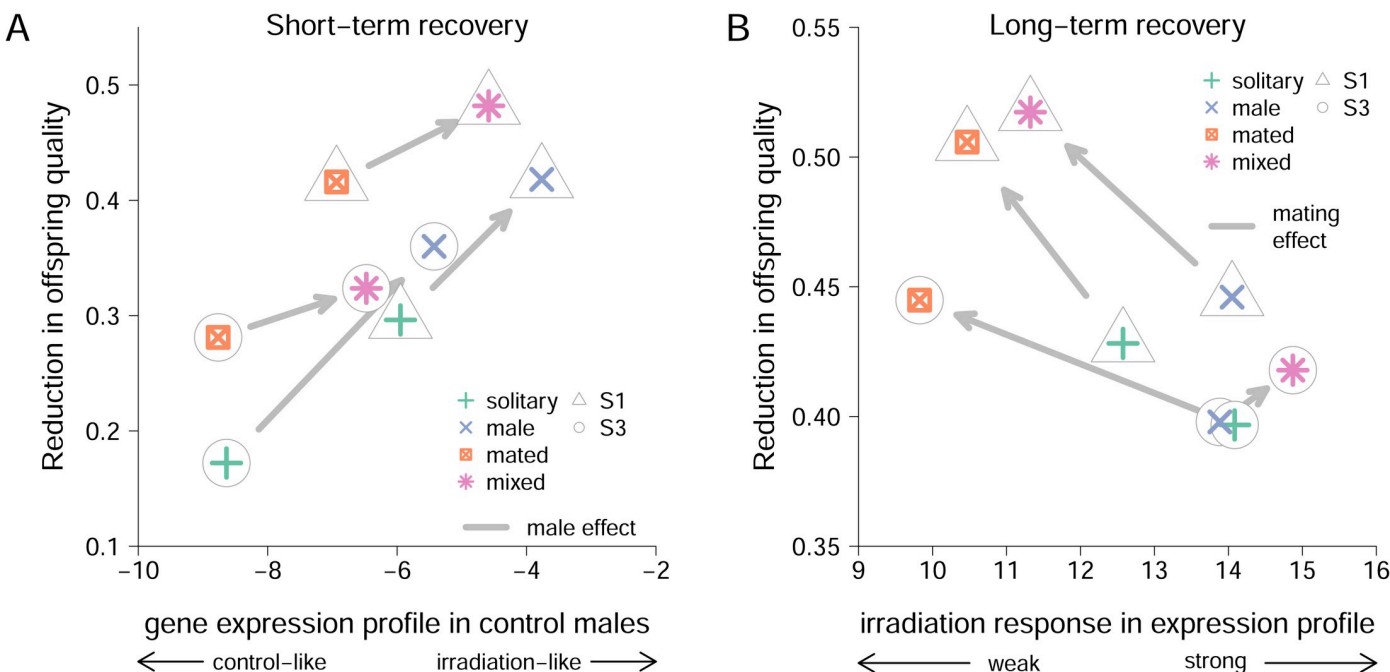

**Fig 5. Relationship between gene expression profiles in fathers and the reduction in the quality of offspring fathered by irradiated males.** (A) As indicated by the first dimension of the canonical correlation analysis (see Fig 4), stronger reduction in offspring quality after a short-term recovery is associated with a more irradiation-like gene expression profile in control males (corresponding to baseline expression in sociosexual treatments before irradiation), with male–male interactions leading to more irradiation-like gene expression profiles and stronger reduction in offspring quality. Arrows indicate the effect of adding males to the sociosexual environment. (B) According to the second canonical dimension, larger gene expression response to irradiation (seen in unmated males) led to a smaller reduction in offspring quality. Arrows indicate the effect of adding females (and thus mating opportunities) to the sociosexual environment. The data underlying this figure can be found in "MainExp_Log2cpm_IrradiationGenes.txt," "QualityReduction_Short_Means.txt," and "QualityReduction_Long_Means.txt" at Mendeley Data (http://dx.doi.org/10.17632/szyb2z8rzk.1).

### Expression of irradiation responsive genes in experimental evolution lines

It has previously been established that males from the different selection regimes differ in their capacity to maintain their germline after induced damage (Fig 2A and Baur and Berger (2020) [7]). We therefore compared the expression of the 18 irradiation responsive genes across all 8 experimental evolution lines. To this end, we analyzed available data that focused on the reproductive tissue. These data were RNA sequences from male and female abdomens from the experimental evolution lines, taken 24 h after a single mating, with females having access to beans and males being held in groups of 5 males during the 24 h period.

We first calculated canonical scores for males from all experimental evolution lines using the canonical coefficients from our previous analysis (Fig 4B). While there was a tendency for S males to have a more irradiation-like gene expression profile than N and N+S males (Fig 6B), we found no significant difference between the 3 experimental evolution regimes (ANOVA: $F_{2,5} = 1.02$, $P = 0.425$), which, in part, may reflect the low statistical power for the test, comparing regimes represented by only 2 or 3 replicate lines with only a single sample per line, and for slightly different tissues (reproductive tracts versus whole abdomens). In contrast, when analyzing sex differences across all 8 replicate lines, all but 2 genes showed a significant differential expression between males and females (S3 Appendix). Genes up-regulated due to irradiation tend to be female-biased, and genes that are down-regulated due to irradiation tend to be male-biased (Fisher's exact test: $P = 0.017$; Fig 6A), indicating that females generally invest more heavily in germline maintenance than males do.

Our data are consistent with a trade-off between germline maintenance and investment in post-copulatory traits conferring advantages in sperm competition. Thus, we were interested in whether any of the irradiation responsive genes might be involved in sperm offense or defense. Therefore, we tested whether male expression of any of the 18 genes correlated with the estimated sperm defense (P1) and offense (P2), averaged across social contexts for a male's first mating, in the 8 experimental evolution lines. After *p*-value correction for multiple testing, 1 gene (*CALMA-C_LOCUS10093*) retained a significant positive correlation with sperm offense (Fig 6C and S3 Appendix). A higher expression of this gene is strongly statistically associated with a higher sperm offense success in *C. maculatus* males, while being down-regulated in response to germline damage. Since the predicted protein for this gene contains a *TFIIS N-terminal* domain [72], it is likely to be a transcription factor and thus may have a regulatory role in mediating the trade-off between investment into sperm competition and germline maintenance.

## Discussion

We hypothesized that male mutation rate and resulting offspring quality is governed by male strategies balancing the competing needs for post-copulatory reproductive success and germline maintenance. We therefore predicted that intense sexual selection coupled with the removal of constraints imposed by natural selection in the S regime would lead to the evolution of increased male reproductive competitiveness at the cost of germline maintenance.

We first confirmed a key expectation under this hypothesis by showing that S males had evolved increased post-copulatory reproductive success (Fig 1). We then showed that previous findings, demonstrating that S males reduce germline maintenance when engaging in socio-sexual interactions [7], are repeatable and that male–male interactions (without mating) can be enough to elicit this response. We sequenced male reproductive tracts and identified 18 candidate genes that show differential expression in response to induced damage in the germline of S males. Many of those genes' predicted protein products contain domains implicating their involvement in cellular maintenance and DNA repair. In line with observations of generally lower germline mutation rates in female animals, we found that genes that were up-

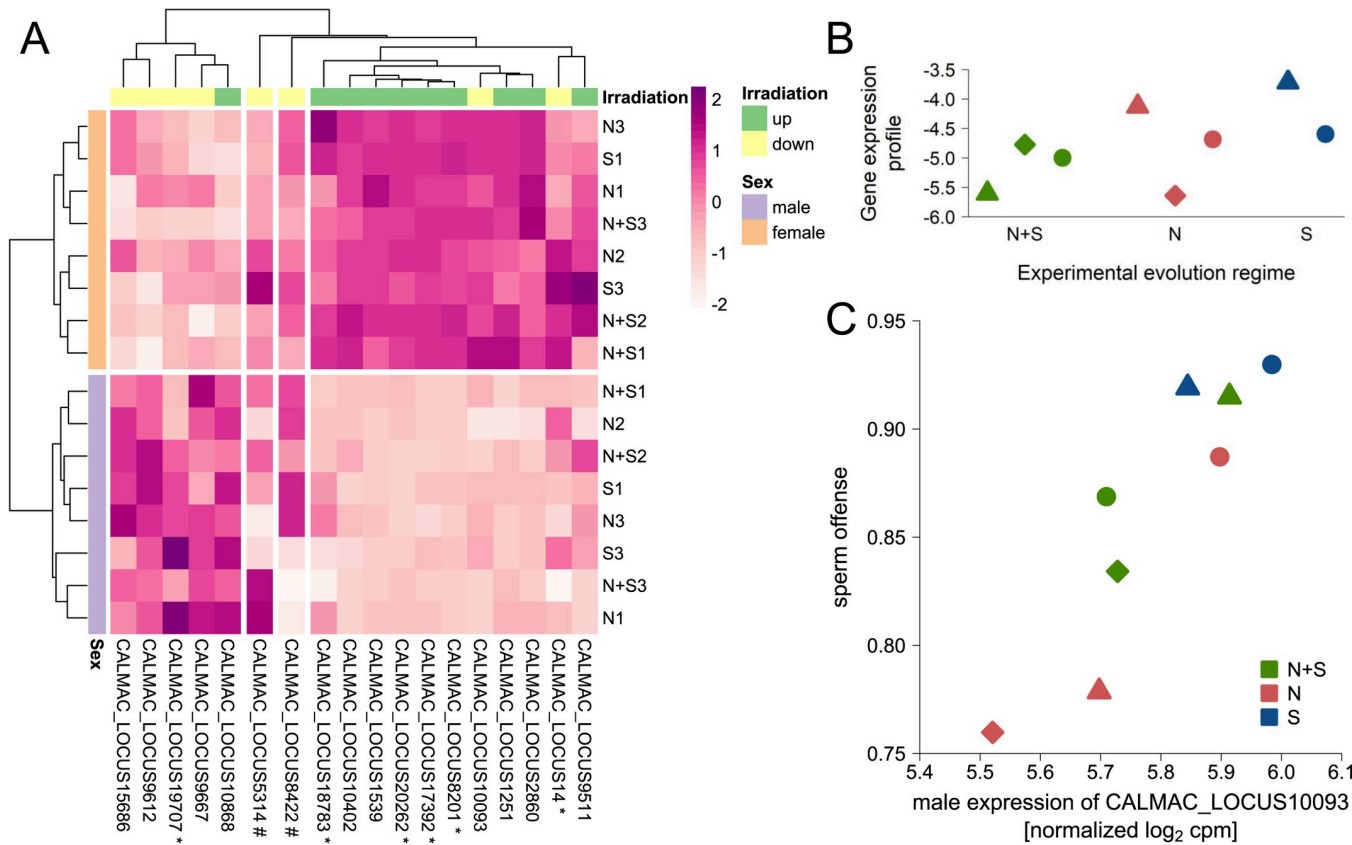

**Fig 6. Expression of the 18 irradiation responsive genes in the 8 experimental evolution lines.** (**A**) Heatmap of scaled normalized log$_2$ expression values. Samples are separated by sex (females: orange; male: purple) and genes are separated by sex bias, which roughly coincides with the direction of irradiation response (up-regulated: green; down-regulated: yellow). All but 2 genes (#) show a significant differential expression between males and females; 6 genes show a significant sex bias of at least 2-fold difference (*). Genes being up-regulated in response to irradiation tend to be female-biased (right block), while genes being down-regulated in response to irradiation tend to be male-biased (left block). (**B**) Scores (based on canonical coefficients used previously to separate control and irradiated samples) of male samples from the experimental evolution lines (24 h after a single mating). Higher scores indicate a more irradiation-like gene expression profile. (**C**) Positive correlation between normalized log$_2$ expression of the irradiation responsive gene *CALMAC_LOCUS10093* (down-regulated in response to germline damage) and sperm offense (P2) ability of males from the experimental evolution lines. Experimental evolution regimes: N +S: natural and sexual selection; N: only natural selection; S: mainly sexual selection. Triangles refer to S1, N1, N+S1; diamonds refer to N2, N+S2; and circles refer to S3, N3, N+S3. The data underlying this figure can be found in "EE_Lines_Log2cpm_IrradiationGenes.txt" and "P2_line_estimates.txt" at Mendeley Data (http://dx.doi.org/10.17632/szyb2z8rzk.1).

regulated in response to germline damage were more expressed in females compared to males, whereas the opposite was true for genes that were down-regulated in response to damage, indicating higher female investment in germline maintenance in *C. maculatus*. The 18 candidate genes also showed significant expression changes across sociosexual treatments in males and mating generally limited their damage response, suggesting that these genes could be involved in a trade-off between germline replication and maintenance. We also found that the expression of these candidate genes in the reproductive tissue of fathers predicted the reduction in quality of their progeny brought about by the induced germline damage. Furthermore, we identified 1 gene whose expression was strongly positively correlated to sperm offense success but down-regulated in response to germline damage, suggesting that the gene could play a role in mediating the trade-off between post-copulatory reproductive success and offspring quality in *C. maculatus*. Our findings thus suggest that changes in the relative strengths of sexual and natural selection can lead to the evolution of phenotypic plasticity in the male reproductive tract with likely consequences for germline mutation rates and offspring quality.

Male reproductive investment is known to be responsive to female characteristics such as mating status [57] and to the presence of conspecific males [52,58], as those cues can indicate the number of future mating opportunities and the level of sperm competition a male can expect [59]. Here, we found that germline maintenance was responsive to the presence of other males even in the absence of females and mating opportunities. This result is also in line with a recent study that found that intense male–male competition increases the fraction of sperm showing signs of DNA damage in zebrafish ejaculates [56]. While the need for investment into reproduction after mating is obvious as males need to replenish sperm and seminal fluid proteins, encounters with rival males do not result in an expenditure of those components and thus may not necessarily stimulate further production of sperm and seminal fluid nor effectuate a reproduction–maintenance trade-off. Yet, encounters with conspecific males can serve as a signal for increased sexual competition that might warrant an increased investment into sperm and seminal fluid proteins that enhance post-copulatory fertilization success. Indeed, in *D. melanogaster*, male–male interactions change a male's gene expression in the somatic (head/thorax) and reproductive (abdomen) tissue leading to the up-regulation of several ejaculate component genes [76], and increasing levels of perceived sperm competition lead to a rise in male seminal fluid protein product and transfer [52].

In addition to strategic allocation decisions in response to conspecific male competitors, it is also possible that the reduction in germline maintenance observed in our study was partly caused by the very costly nature of male–male interactions in *C. maculatus* [70]. In this scenario, manifestation of the trade-off between investment in reproduction (via engagement in reproductive competition) and germline maintenance could result from male–male competition depleting resources overall or shifting them away from the germline to the soma [9,44]. This mechanism could also explain some discrepancies in the finer details of our results linking plastic responses in sperm competition success and germline maintenance. Most centrally, while we both here and previously [7] found that S males have evolved a plastic reduction in germline maintenance in response to social cues, we did not find that their success in sperm competition was improved by such cues (as expected in the trade-off scenario), nor that their response to these cues in terms of sperm competition success was much different from that of the other regimes (although low statistical power may have played a role here). This, however, could be explained by male–male interaction being overall very detrimental to male condition [70], reducing sperm competition success. S males tend to engage more in such costly interactions than N and, possibly, also N+S males [7]; therefore, because males were co-reared with males from their own regime in our experiments, it is possible that the detrimental effects of male cohabitation were most pronounced for S males in our assays of sperm competition success. Hence, even though there is differential allocation to sperm traits in response to social cues in S males compared to N and N+S males [7], any positive effects from this on sperm competition success may have been masked by stronger negative effects of male cohabitation in the S regime in our experiment. Interestingly, the decreased sperm defense (P1) seen in competing S males was not observed for sperm offense (P2), for which S males were superior (Fig 1), which indeed hints at changes in ejaculate allocation/composition in response to social cues.

Our gene expression data offer potential mechanistic insights into the allocation trade-off between sperm competition success and germline maintenance. In theory, the observed reductions in offspring quality fathered by males engaging in sociosexual interactions could result from an increase in sperm production while keeping maintenance constant, rendering more replication errors unchecked per gamete. While this would not represent a functional allocation trade-off between maintenance and reproduction, it would still result in a trade-off between male success in sperm competition (assuming that success is dependent on sperm

numbers) and gamete quality. Importantly, however, our gene expression data indicate that males engaging in mating interactions also have a decreased capacity to respond to DNA damage (Fig 5B). Indeed, this result is in line with previous findings from these lines showing that reductions in offspring quality under inflated levels of germline damage is not a simple function of either sperm age or sperm production rate [7]. This previous study showed that sociosexual treatment, the largest contributor to sperm replenishment and thus sperm age, had no general effect on offspring quality reduction, but rather only affected S males [7]. Furthermore, ejaculate investment patterns did not indicate a higher amount of sperm and seminal fluid being transferred by S males in an intrasexual competition setting compared to S males reared in isolation, which makes sperm production rate an unlikely explanation for the observed reduction in offspring quality [7].

Our gene expression data also suggest that *C. maculatus* males devote less resources into germline maintenance than their female counterparts. Given that sexual selection acts more strongly in males in most species [77,78], including *C. maculatus* [79,80], a trade-off between germline maintenance and post-copulatory reproductive success thus offers an explanation for the widely observed male mutation bias that goes beyond invoking sex differences in the numbers of germline replications as the sole driver of the phenomenon [7,38,40]. Data from other species are scarce, but there is some correlative comparative evidence to support a trade-off between germline mutation rate and post-copulatory reproductive investment. For example, testes mass (a possible adaptation to sperm competition) and DNA substitution rates have been shown to covary in primates [81], and across bird taxa estimates of mutation rate have been shown to correlate with extra-pair paternity (as a measure of sperm competition intensity), but notably not with relative testes mass [39]. However, if these correlative patterns are indeed causal, and what their mechanistic explanation may be, remains unknown.

In *C. maculatus*, males that are successful in sperm offense sire daughters with lower lifetime reproductive success [82], which accords with the presence of substantial amounts of sexually antagonistic genetic variation for fitness in this species [83]. However, our study suggests that this effect may in part be mediated through reduced germline maintenance in successful males leading to lower genetic quality of their offspring. If so, a similar reduction in quality would also be expected for sons of successful males (which remains unconfirmed). This in turn may steer female choice away from these males in order to enhance the genetic quality of offspring [84]. However, systematic differences in male and female expression of genes involved in germline maintenance, as indicated here, and the capacity of females to repair damaged male sperm [6,85,86] opens up for the possibility that females instead may devote large amounts of resources to maintenance of ejaculates and sperm deriving from males that are superior in sperm competition. Thus, male–female coevolution of germline maintenance may represent an important, yet understudied, aspect in the evolution of mate choice. More generally, the evolution of phenotypic plasticity in germline maintenance in response to sexual competition, as demonstrated here by experimental manipulation, might contribute to both systematic differences in mutation rate between the sexes as well as among-male variation within species. Such variation has fundamental implications for theories of sexual selection, including reinforcement of mate choice and "good genes" processes, and for the maintenance of genetic variation in fitness related traits.

## Methods

### Experimental evolution lines

All beetles for the experiments were reared and kept on black-eyed beans (*Vigna unguiculata*) in constant climate chambers at 29°C, 50% relative humidity, and a 12:12 L:D cycle. Focal

individuals came from 8 experimental evolution lines that originated from the *Lomé* popula-
tion and are described in detail in [7,68]. The lines had evolved for >50 generations under one
of three experimental evolution regimes: N+S beetles evolved under polygamy with opportuni-
ties for natural (N) and sexual (S) selection to act, N beetles evolved under enforced monog-
amy with sexual competition between males removed and thus mainly natural (N) selection
acting, S beetles evolved under polygamy but with a middle-class neighborhood breeding
design applied to females weakening natural selection and leaving mainly sexual (S) selection
to act. In brief, sexual selection in S males was imposed by allowing all males from a given line
to compete over matings with their conspecific females. After mating and competition, females
were isolated individually and allowed to lay eggs. We allowed each female to only contribute a
single son and daughter to the next generation. This thus relaxed selection on female fecundity
in the adult stage while allowing for intense sexual selection (see also [66,68,87]). For the germ-
line maintenance experiment, we focused on males from the 2 S lines, which were both mated
to females from a third (polygamous) line to exclude any female derived and/or coevolution
effects. For the sperm competition experiment and the second gene expression dataset, we
included beetles from all 8 experimental evolution lines.

## Sperm competition

Males from all 8 experimental evolution lines were tested for the sperm competitiveness when
being first (sperm defense, P1) or second (sperm offense, P2) to mate with a female that was
mated to a competitor male from a black strain of *C. maculatus* [69] after or before the focal
male, respectively. To avoid potential confounding effects of cryptic female choice, we used
females from the ancestral *Lomé* population (see [84,88] that were mated to 2 males (observed
single matings) 24 hours apart. Focal males were held in one of 2 social environments for
approximately 24 h before their mating: solitary (single males in 30 mm dishes) or competition
(in groups of 5 males in 90 mm dishes). The experiment was conducted over 2 temporal
blocks. Within each of the 2 blocks, lines were separated into one of 3 sub-blocks, 2 sub-blocks
contained 1 line from each of the 3 experimental evolution regimes and the last block con-
tained the third replicate line of the N+S and N regime. In each block virgin males and females
from the experimental evolution lines, the black competitor line and the ancestral stock popu-
lation were collected within 24 h after eclosion. Experimental evolution line males were imme-
diately transferred into one of the 2 social environments; black males were all held in groups of
up to 10 males until their mating (that took place either on the same day or the day after), thus
keeping male age and mating status (virgin) constant for all males. Similarly, females from the
ancestral population were held in groups of up to 20 virgin females until their first mating
(that took place 1 to 2 days after collection).

For sperm defense, females were mated to a focal male from one of the experimental evolu-
tion lines in a single observed mating and afterwards kept on beans for 24 h before their sec-
ond mating. Afterwards, all females were given the opportunity to mate with a black
competitor male within 40 min. Successful mating pairs were separated after the end of the
mating, and females were moved onto fresh beans for 48 h of egg laying ($N = 290$, with a mini-
mum of $N = 26$ per line). Beans with eggs were incubated for approximately 30 days, and
emerged beetles were frozen at −20°C. Afterwards, offspring were counted and separated by
black and wild-type fathers based on their coloration.

For sperm offense, stock females were first mated to a black male in observed single matings
and kept on beans for egg laying in groups of max. 25 for 24 h. Afterwards, females were given
the opportunity to mate with 1 focal male from the experimental evolution lines for 40 min.
While this was always the second and last mating for the female, the mating represented one of

5 consecutive matings for the focal male. Females from a focal male's first, third, and fifth mating were put on fresh beans for 48 h to determine focal male sperm offense success ($N = 784$, with a minimum of $N = 17$ per line and mating). Beans were incubated for approximately 30 days, and emerged beetles were frozen at −20°C. Afterwards, offspring were counted and separated by black and wild-type fathers based on their coloration.

Statistical analyses and preparation of graphs were done in R 4.1.1 [89] using Bayesian Generalized Linear Mixed Effect Models implemented within the package *MCMCglmm* [90]. Proportion of the focal male's offspring was modelled with binomial error distribution corrected for overdispersion. When analyzing P1 (sperm defense), we included experimental evolution regime as well as its interaction with the social treatment as fixed effects. We modelled variance between experimental evolution lines per social treatment, as well as experimental (sub)blocks, as random terms. When analyzing P2 (sperm offense), we additionally included mating number and its two-way interactions with experimental evolution regime and social treatment as fixed effects. However, all interactions between evolution regime and mating number were nonsignificant and removed from final models to ease interpretation (see S1 Appendix). Again, we modelled variance between experimental evolution lines per social treatment but also added a crossed random term capturing variation in how lines responded to mating number. As additional random terms, we included experimental block effects specific to the social environment and male ID. When analyzing total success in sperm competition (P1 + P2), we modelled the main effects of evolution regime while adding the additional fixed terms of "paternity measure" (two-level factor: P1 or P2), mating number, and social treatment. We included evolution line, male ID, and block effects (specific to social treatment and paternity measurement) as random terms. We used weak and uninformative inverse-Wishart priors [90] for all random effects in all models. To retrieve Bayesian *P* values and 95% credible intervals, we calculated marginal means from the stored posterior distributions. We performed pairwise contrasts focusing specifically on differences between the S regime and the other 2 regimes. For example, to compare P1 between S and N males and calculate the Bayesian *P* value for whether the 2 regimes differed overall, we calculated the marginal means for S and N males by averaging P1 across the 2 social treatments in each stored posterior. The two-sided *P* value was then calculated as the fraction of posteriors in which the regime with the lowest P1 on average had higher P1, multiplied by a factor of 2. Model specification and output for all 3 response traits (P1, P2, and Total success) are reported in full in S1 Appendix.

### Germline maintenance

**Experimental assay.** In order to measure germline maintenance capacity, we induced DNA damage in adult males by exposing them to 25 Gray of gamma radiation (for 35 min at a dose rate of 0.72 Gray/min from a cesium-137 source). Gamma radiation causes double- and single-strand DNA breaks as well as increases the amount of reactive oxygen species in cells [34], which, in turn, can induce further DNA damage [34,45]. While our treatment drastically increased germline damage, DNA breaks occur naturally during both recombination and chromatin remodeling during sperm development, and errors in the repair of those breaks give rise to point mutations [29,34,91]. The number of DNA lesions induced by a given dose of gamma radiation is surprisingly constant per DNA base pair [91], and, thus, differences in mutation rate are mainly caused by to the amount and type of repair molecules [34,91,92], which makes this assay ideal for measuring germline maintenance. Because most mutations are neutral or deleterious [93,94], the amount of mutations transferred from parents to offspring can be approximated by the decline in offspring quality of parents that were challenged to deal with elevated levels of reactive oxygen species and DNA damage [6–8,10,15].

Assays were replicated on 2 consecutive days. Males and females (from the 2 S lines) and females (from a third, polygamous line) were picked as virgins within 24 h after emergence. Females (from the third, polygamous line) were held in groups of 10 in Petri dishes (90 mm) until mating assays and males were immediately transferred to their respective sociosexual environment using females from their own experimental evolution line where applicable. Sociosexual treatments were set up by manipulating the presence of conspecific males and females in a full factorial design. Males were held in a 35-mm Petri dish without any conspecifics or with a single virgin female, or in a 90-mm dish together with 4 conspecific males or with 4 conspecific males and an additional 5 virgin females. Males were held in their respective sociosexual environment for approximately 24 h until shortly (<1 h) before the irradiation treatment. Then, males were separated into individual 0.5 ml reaction tubes with a hole punched into the lid. Roughly half of the males then underwent a radiation treatment, while the other half served as controls.

**Offspring quality.** Germline maintenance was assessed by measuring fitness effects of the induced germline damage in subsequent generations. Shortly after irradiation (1.5 to 3 h day 1, 2.5 to 4 h day 2), males were mated once to a single virgin female (0 to 48 h old) in a 60-mm Petri dish on a heating plate set to 29˚C. Females were put on beans to lay eggs for 72 h, and males remained in their individual Petri dishes to renew their ejaculate, thus making sure that all males were challenged to deal with the competing tasks of both replicating and maintaining their germline. One day after irradiation (22 to 24 h day 1, 22 to 23 h day 2), males were again mated to a single virgin female (24 to 48 h old) in 60 mm dishes on a heating plate. Females were put on beans for 72 h to lay eggs, and males were discarded. All beans were incubated at 29˚C, 50% r. h., and 12:12 L:D cycle in a climate chamber for 30 days to ensure that all viable offspring had emerged. Before offspring eclosion, beans were transferred to virgin chambers so that virgin $F_1$ offspring could be collected for assaying offspring quality.

To determine the reduction in quality of offspring fathered by irradiated males, we crossed the $F_1$ offspring of each male with the $F_1$ offspring of other males within the same treatment, experimental evolution line, and experimental day using a Middle-Class Neighbourhood breeding design (relaxing selection on the induced mutations by attempting to equalize the contribution of each $F_0$ male to the $F_1$ progeny used for further assaying; see [87]). For the first ejaculate, we aimed at crossing 1 $F_1$ male and 1 $F_1$ female per $F_0$ male (resulting in a total of 387 assayed $F_1$ couples). For $F_1$ offspring deriving from the $F_0$ male's second ejaculate, we aimed at crossing 3 males and 3 females (resulting in a total of 1,172 assayed $F_1$ couples), as we wished to focus on the recovery of the male germline. Pairs were kept on beans for their entire life, and we incubated dishes for 33 days before freezing them at −20˚C to count $F_2$ offspring production. Counts of $F_2$ adults emerging from these irradiated lineages (n $F_0$ males = 224) were compared to counts from corresponding control lineages (n $F_0$ males = 163) to calculate reduction in offspring quality as: 1-[F2$_{IRRADIATED}$/F2$_{CONTROL}$]. Thus, we could explore phenotypic plasticity in germline maintenance in response to the sociosexual treatments by comparing reduction in quality of offspring from males kept under the 4 treatments (Fig 2B).

We used Bayesian Generalized Linear Models implemented within the package *MCMCglmm* [90] in R 4.1.1 [89] for statistical analyses. Number $F_2$ offspring were modelled with Poisson error distribution corrected for overdispersion, with dam and sire (IDs of the 2 grandfathers) entered as a multiple-membership random term. The fixed effects of sociosexual interactions were modelled as 2 two-level factors (Inter- and Intrasexual interactions) testing for a significant interaction with irradiation treatment. We also added experimental evolution line and day as fixed effects to test for any differences between the 2 lines and the 2 assay days. To ease interpretation, nonsignificant interaction terms were removed (both full and final reduced models presented in S2 Appendix). For graphical presentation, line-specific means

(and their 95% Bayesian credible intervals) of the reduction in offspring quality of irradiated males relative to control males were calculated per sociosexual environment based on the posterior estimates from models equivalent to those specified above, but using a Gaussian distribution for the response variables. Packages *Hmisc* [95] and *RColorBrewer* [96] were used to generate graphs.

**Differential gene expression.** For RNA extraction, beetles were snap frozen 2 h after irradiation treatment and stored at −80°C until dissections. During dissections, beetles and dissected tissues were kept on dry ice and afterwards stored at −80°C until RNA extraction. Males were dissected on ice in a droplet of PBS; the entire reproductive tract (Fig 2C) was removed; and the two large accessory gland pairs cut off. We decided to remove the two large accessory gland pairs in order to keep dissections consistent, as the large accessory glands easily detach and/or rupture during dissections. Afterwards, the remaining tissue (aedeagus, ejaculatory bulb, two bilobed testes, and three pairs of smaller accessory glands [ectadenial glands] [97]) was quickly rinsed in a fresh droplet of PBS and then transferred to a reaction tube on dry ice. We aimed to pool tissue from 10 males per sample; for 2 samples (1 mated irradiated line S3 and 1 mated control line S3), we only obtained tissue from 9 males. Each sample consisted only of males from the same treatment, line, and experimental day. This resulted in a total of 32 samples with 4 replicates per treatment (1 per day and line).

RNA was extracted with Qiagen RNeasy Mini Kit, and on-column DNA digestion was performed with Qiagen RNase free DNase Kit. We followed the manufacturer's instructions; beta-mercapto-Ethanol was added to the lysis buffer, and tissue lysis was done with 1 stainless steel bead in a bead mill at 28 Hz for 90 s. Two samples underwent an additional cleanup using the Qiagen RNeasy Mini Kit. RNA concentration and purity were assessed with NanoDrop, and additional quality controls were performed at the sequencing facility. Samples were sequenced at the SNP&SEQ Technology Platform in Uppsala. Libraries were multiplexed and sequenced as stranded paired-end 50 bp reads in 2 lanes of a NovaSeq SP flow cell resulting in roughly 11 M to 26 M reads per sample.

Raw reads were inspected with *FastQC* [98] and quality information summarized with *MultiQC* [99]. We then mapped all reads to the *C. maculatus* genome (GCA_900659725.1; ASM90065972v1) [100] with *TopHat 2.1.1* [101] allowing for up to 2 mismatches per read and including strand information. We only kept reads where both mates successfully mapped to the *C. maculatus* genome. Those reads were then counted per gene using *HTSeq* [102] with default settings for stranded libraries. Statistical analyses were done in R 4.1.1 [89], with packages *edgeR* [103] and *limma* [104]. Libraries from the 2 lanes were merged into 1 sample. Genes that were not at least expressed as 1 count per million (cpm) in at least 2 samples were excluded from the analysis resulting in a total of 12,161 genes being analyzed. Counts were normalized with the "Trimmed Mean of M-values" method and normalized $\log_2$ cpm values were analyzed in linear models within *limma* [104]. Experimental evolution line and experimental day were added as additive terms to control for variance between lines and days. Sociosexual environment was entered as a 4-level factor and irradiation treatment as a 2-level factor. Due to the low number of genes responding to the irradiation treatment, we lacked statistical power to analyze the interaction between social environment and irradiation with the full set of genes. Therefore, the interaction was removed from the model, and we analyzed the interaction in a separate model considering only genes that responded to the irradiation treatment. We used information available on *UniProt* [72] accessed on 10.03.2022 to gain insight on the potential function of some of the genes found to be differentially expressed.

For further analyses, we always used normalized $\log_2$ cpm values. Using the 18 irradiation responsive genes, we ran a multivariate ANOVA. Additionally, we ran a linear discriminant analysis on gene expression in the 18 irradiation responsive genes to find a linear combination of the

expression of these genes that best separates irradiated from control samples. To this end, we controlled for variation arising through differences in irradiation response between lines, sociosexual environments, or experimental days using package *candisc* [105]. To avoid overfitting the data, we calculated canonical scores of the 32 samples with the first canonical axis only. For further analyses and graphical representation, we used mean canonical scores across the 2 experimental days. To estimate how well differences in gene expression correspond to differences in reduction in offspring quality due to germline damage, we conducted a Canonical Correlation Analysis. We added canonical scores of irradiated and control samples (averaged across the 2 experimental days) as two x variables and reduction in offspring quality after a short- and long-term recovery period as two y variables to the Canonical Correlation Analysis implemented in the package *CCA* [106]. For graphical presentation, heatmaps were constructed on scaled normalized $\log_2$ cpm values using hierarchical clustering and Manhattan distance metrics in *pheatmap* [107], and additional packages *VennDiagram* [108] and RColorBrewer [96] were used.

## Expression of irradiation responsive genes in experimental evolution lines

To analyze the expression of irradiation responsive genes in the 8 lines from all 3 experimental evolution regimes, we made use of an existing data set designed to study the evolution of sex-biased gene expression under these selection regimes. Before collecting individuals for sequencing, all experimental evolution lines underwent 3 generations of common garden rearing (i.e., a polygamous mating setting). Males and females from the experimental evolution lines were mated once (observed) to a standardized mating partner from the *Lomé* population on heating plates set to 29˚C. Matings were separated into 4 blocks, and in each block, we set up 6 mating pairs per line and sex. Beetles from the first 5 successful matings per line and sex were separated after the end of the mating; focal females were kept singly on beans for 24 h, and focal males were held together in a 90-mm dish (in groups of 5 individuals) for 24 h. After 24 h, males and females were flash frozen in liquid nitrogen and kept at −80˚C until sample preparation. Since we were interested in the reproductive tissues, we only sampled the abdomen of males and females. To that end, we separated the abdomen from the rest of the body on ice, while storing samples on dry ice during preparation. We pooled 12 abdomen per group balanced over the 4 blocks (except for the male sample from line N+S2, which only contained 10 abdomen; block information on the 2 lost abdomen is not available). After dissection, samples were stored again at −80˚C until RNA extraction.

RNA was extracted with Qiagen RNeasy Mini Kit and on-column DNA digestion was performed with Qiagen RNase free DNase Kit. We followed the manufacturer's instructions; beta-mercapto-Ethanol was added to the lysis buffer, and tissue lysis was done with 2 stainless steel beads in a bead mill at 28 Hz for 90 s in 700 µl lysis buffer. After centrifugation, the entire supernatant was transferred to a fresh tube, mixed quickly, and 350 µl went onto the extraction column while the rest was discarded to avoid overloading the column. RNA was eluted 2 times in 50 µl of RNase-free water each. RNA concentration and purity were assessed with Nano-Drop, gel electrophoresis, and Qbit; additional quality controls were performed at the sequencing facility. Samples were sequenced at the SNP&SEQ Technology Platform in Uppsala. Libraries were multiplexed and sequenced as stranded paired-end 150 bp reads in 1 lane of a NovaSeq S4 flow cell resulting in roughly 24 M to 56 M reads per sample.

Gene counts were obtained and analyzed as in the main experiment with the exception of an additional quality and adapter trimming step with *Trimmomatic* [109] and allowing for up to 8 mismatches per read during mapping due to the higher read length. Normalized $\log_2$ cpm counts of all 12,874 retained genes were analyzed in a linear model within *limma* [104]. We constructed an additive model with sex (2-level factor) and experimental evolution regime

(3-level factor) as explanatory variables. Sex bias was estimated across all experimental evolution regimes (male–female), and $p$-values were corrected with Benjamini–Hochberg method using a 5% FDR cutoff. We then extracted normalized $\log_2$ cpm values of the 18 irradiation responsive genes for all samples for further analysis. Using these values, we predicted canonical scores for males from all experimental evolution lines based on the linear coefficients from the previous analysis.

## Supporting information

**S1 Appendix. Sperm competition results.**
(DOCX)

**S2 Appendix. Offspring quality.**
(DOCX)

**S3 Appendix. Gene expression.**
(DOCX)

## Acknowledgments

We thank P. E. Eady for sharing the *C. maculatus* black line with us and J. Liljestrand-Rönn for help in the lab. Sequencing was performed by the SNP&SEQ Technology Platform in Uppsala. The facility is part of the National Genomics Infrastructure (NGI) Sweden and Science for Life Laboratory. The SNP&SEQ Platform is also supported by the *Swedish Research Council* and the *Knut and Alice Wallenberg Foundation*.

## Author Contributions

**Conceptualization:** Mareike Koppik, David Berger.

**Data curation:** Mareike Koppik, Julian Baur.

**Formal analysis:** Mareike Koppik, David Berger.

**Funding acquisition:** David Berger.

**Investigation:** Mareike Koppik, Julian Baur, David Berger.

**Methodology:** Mareike Koppik, Julian Baur, David Berger.

**Project administration:** David Berger.

**Supervision:** David Berger.

**Visualization:** Mareike Koppik.

**Writing – original draft:** Mareike Koppik, David Berger.

**Writing – review & editing:** Mareike Koppik, Julian Baur, David Berger.

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
