## [Editor Report · Decision Letter 0]

26 Oct 2022

Dear Dr Berger, 

Thank you for submitting your manuscript entitled "Increased male investment in sperm competition results in reduced maintenance of gametes" for consideration as a Research Article by PLOS Biology.

Your manuscript has now been evaluated by the PLOS Biology editorial staff as well as by an academic editor with relevant expertise and I am writing to let you know that we would like to send your submission out for external peer review.

Once your full submission is complete, your paper will undergo a series of checks in preparation for peer review. After your manuscript has passed the checks it will be sent out for review. To provide the metadata for your submission, please Login to Editorial Manager (https://www.editorialmanager.com/pbiology) within two working days, i.e. by Oct 28 2022 11:59PM.

Kind regards,

Lucas

Lucas Smith, Ph.D.

Associate Editor

PLOS Biology

lsmith@plos.org

---

## [Decision Letter · Decision Letter 1]

5 Jan 2023

Dear Dr Berger,

Thank you for your patience while your manuscript "Increased male investment in sperm competition results in reduced maintenance of gametes" went through peer-review at PLOS Biology. Please accept our apologies for the delay in sending you a decision, which was caused, in part, by the Christmas holidays. Your manuscript has now been evaluated by the PLOS Biology editors, an Academic Editor with relevant expertise, and by several independent reviewers.

As you will see in the reviews, which you will find at the end of this email, the reviewers are enthusiastic about your study and they appreciate the interest of your findings to the field. However, they have also raised a number of important issues which need to be carefully addressed before we can consider your manuscript for publication. We are therefore pleased to offer you the opportunity to address the comments from the reviewers in a revision that we anticipate should not take you very long. We will then assess your revised manuscript and your response to the reviewers' comments with our Academic Editor aiming to avoid further rounds of peer-review, although we might need to consult with the reviewers, depending on the nature of the revisions.

**In addition to addressing the reviewer comments, when revising your study we also ask that you attend to the following editorial request:

1) DATA REQUEST: Thank you for providing the data underlying your study as depositions to the GEO and Mendeley Data repositories. We ask that you please add a sentence to each figure legend directing readers to where this data can be found. For example, to each figure legend (including supplemental), you could add the sentence "the data underlying this figure can be found in ___".

**IMPORTANT - SUBMITTING YOUR REVISION**

*Resubmission Checklist*

*Published Peer Review*

*PLOS Data Policy*

*Blot and Gel Data Policy*

Sincerely,

Luke

Lucas Smith, Ph.D.

Associate Editor

PLOS Biology

lsmith@plos.org

REVIEWS:

Reviewer #1: In a multi-step experiment using experimental evolution lines (natural versus sexual selection), irradiation, and phenotypic and transcriptomic measurements, the authors found and confirmed a potential trade-off between male investments in sperm competition (via germline replication rate) and offspring quality (via germline mutation rate). 

This is a great study that will be relevant to the field. If found the study to be rigorous and carefully conducted, and the paper reads well. I do not have any major criticism but suggest addressing a few minor issues:

L20 male bias (not hyphenated, though 'male-biased …' often would be)

L38 also here: sex differences 

L66 I think this link to sperm competition success should be made more explicit. I agree that males may have a higher germline mutation rate than females, and it seems clear that higher germline replication rates can generate more sperm, which is indeed under selection through sperm competition. But these comparisons are either between sexes or (mostly) between species (with species that have higher levels of sperm competition producing more sperm and possibly at a higher rate). Translating such arguments to variation within a natural population of a single species (where sperm competition occurs and offspring quality is under selection), however, seems a stretch to me. Is there any good evidence to suggest that some males replicate their germline at a substantially higher rate than others of the same population? Variation in mutation rates between males I can see, but more often, I would argue, differential sperm production stems from variation in testes size (accommodating fewer or more germlines in parallel) but not necessarily from different rates of cell division. Can you please clarify? 

L69 division -> divisions

L156 How did you control for mating activity in the treatments with mating partners? This would be good to mention here as the potential to mate and actually copulating is not the same here and could influence the interpretation of the results. 

L171 In Bayesian statistics, the error bar around posterior means is typically referred to as 95% credible interval, not 95% confidence interval

L183 found instead of 'find' for consistency in tense

L186 One of the key points of Bayesian statistics is to move away from P-values and significance thresholds, but instead focus more on the 'degree of belief' in an event. Reporting P-values and interpreting them as 'marginally non-significant' thus seems to be at odds with the Bayesian framework.

L207 DEG should be defined at first mention - it is widely known, but not everyone is familiar with transcriptomics terminology

L368 damage response is not hyphenated

L380 in -> is (before known)

L395/411 space missing before citation

L564 (and throughout) Petri dish is usually capitalized (it's a name)

L578f It might be helpful to briefly explain Middle-Class Neighbourhood breeding design here.

Reviewer #2: This is an outstanding study that will surely impact the field in a very positive way, and that provides a leap in our understanding of reproductive trade-offs, male-female coevolution and sexual selection at large. In this paper, the authors combine a set of behavioural and fitness assays using beetles from experimentally evolved lines with gene-expression studies and tests of germline damage and maintenance. The main aim is to provide causal evidence that sperm competition may favour increased male germline replication rates, but at the cost of greater germline mutation rates and thus germline maintenance and repair. Their dataset provides solid and consistent evidence that this is the case, all the more impressive because results from different empirical approaches combine and complement each other beautifully. All in all, this study provides tantalizing evidence in favour of an idea that could potentially explain the widespread sex-bias in germline mutation rates observed across the tree of life. In this sense, this study will very likely open up a novel avenue of research. This is the type of study you review and immediately gets you thinking about interpretation of your own results and implications for future work. The design of experiments is sound, the analyses are well-grounded, and the manuscript is very well written, with a very nice introduction and a nuanced and balanced discussion. As a cautionary note, I am not an expert on gene-expression studies, so I can't provide informed feedback on the methods used to this regard. Below I list a series of minor suggestions with the aim of improving an already excellent piece of work. My sincere congratulations to the authors.

Line 40. I suggest re-writing this part. "The choice of an individual" is ambiguous and can lead to misunderstandings that can be easily avoided.

Line 123. The authors should briefly discuss what the consequences of having just two replicate EE lines may have for the interpretation of their results. Three EE replicates is already on the low side of EE replication and thus it seems some justification here is warranted. 

Line 167. Here and in the legend for figure 3 the use of "virgin" is awkward and misleading. These males were only virgin during one part of the assays (obviously, as virgin males cannot have offspring). I suggest re-wording to "solitary" or "isolated". 

Lines 360-378. I miss here a sentence about how upregulated genes in females overlap with up/downregulated genes in response to radiation. This is a very nice result and should be highlighted here too. It also fits very nicely with the overarching idea of the study. 

Line 380. "is" instead of "in" before "known".

Line 396. Cite Hopkins et al. 2019 PNAS here too, it is a very relevant paper that also backs up what you're stating here ("Divergent allocation of sperm and the seminal 

proteome along a competition gradient in Drosophila melanogaster"). 

Line 466. Perhaps delete the comma after "both"?

Line 482. Explain or provide a suitable reference. 

Line 491. This seems to imply females mated three times overall. Please re-write to avoid confusion. 

Line 501. You mean emerged beetles here, but the wording implies you're freezing beans. 

Line 504. Clarify by explicitly stating that pairs were separated after the end of matings. 

Line 512. Explicitly state that males mated with different females, so that each female only mated twice. This is implied but it needs to be unambiguously stated to avoid misunderstandings (if males mated repeatedly with the same female this would conflate other effects). 

Line 596. Spell out HPD.

Reviewer #3: Overall I thought this was an interesting paper. In general it is well written and the results of broad significance

I have a few comments that are quite minor - but nonetheless quite important. My main concern relates to the level of detail around the sperm competition assays (and how they were analysed). The reader needs to know (have confidence) that there was some control over the age and mating status of the focal males, the competitor males and the females. This is important as male age and mating status can affect ejaculatory traits in this species. How the data were analysed was somewhat confusing to me. It does all look rather complicated, for what is fundamentally a quite simple study. It is not clear to me how the model (S2) actually works. I am unsure how a model of 'sperm competition success' can have a fixed effect of number of matings, when this applies (as far as I can tell) to the sperm offense part of 'sperm competition success'? Why not analyse sperm defense and sperm offense separately? 

Line 133: I think it is misleading to suggest a result in which p=0.078 is a strong tendency for higher success. I have similar issues with the marginal approach of a p = 0.088 on lines 184 to 186. I appreciate you use the caveat marginally non-significant, but the essence of meaning can be promoted via the use of terms such as strong tendency or by actually saying there was a decrease in offspring quality. Mating did not decrease offspring quality after the 3h recovery period - your study failed to reject the null-hypothesis. I would prefer a more conservative approach when dealing with violations of statistical orthodoxy.

Line 135: p-value? By providing p-values the reader can decide on marginal v non-marginal calls. 

line 136: p-value? 

Line 137: Did prior matings affect success in sperm competition? 

Line 137: " Similarly, the sperm competition advantage of S males was not significantly different for offense and defense" -what exactly does 'sperm competition advantage' mean? The P1 and P2 values are very different - how do I get the ns result from looking at the table in S2? It is not very clear

line 162: Figure 2 - it seems unusual to have fig 2A, whch presents data from Baur & Berger 2020? Fig 2 is also a little misleading in that it looks as if the quality of F1 offspring (and not F2) are being assayed for quality.

line 250: 'revealed differences between social environments' - did it? They look remarkably similar in fig 4C. Based on what statistical evidence? 

line 362: could this be some type of paternal effect via nutrients transferred within the ejaculate? Being in a socio-sexual environment for 24h (quite a long time for an insect that lives for about a week or so as an adult) will use up resources, which will detract from investment in the ejaculate (reduced size, reduced nutrients, reduced water content) - all of which females may use to invest in offspring

---

## [Editor Report · Decision Letter 2]

22 Feb 2023

Dear Dr Berger,

Thank you for the submission of your revised Research Article "Increased male investment in sperm competition results in reduced maintenance of gametes" for publication in PLOS Biology. Your revision has now been evaluated by the PLOS Biology editorial staff and by the Academic Editor, and we are satisfied by the changes made and feel the revision has adequately addressed the reviewer comments. Therefore, on behalf of my colleagues and the Academic Editor, Michael D Jennions, I am pleased to say that we can in principle accept your manuscript for publication, provided you address any remaining formatting and reporting issues. These will be detailed in an email you should receive within 2-3 business days from our colleagues in the journal operations team; no action is required from you until then. Please note that we will not be able to formally accept your manuscript and schedule it for publication until you have completed any requested changes.

PRESS

Sincerely, 

Lucas Smith, Ph.D.

Associate Editor

PLOS Biology

lsmith@plos.org